# Could a Legume–Switchgrass Sod-Seeding System Increase Forage Productivity?

**DOI:** 10.3390/plants11212970

**Published:** 2022-11-03

**Authors:** Kyriakos D. Giannoulis, Dimitrios Bartzialis, Elpiniki Skoufogianni, Ippolitos Gintsioudis, Nicholaos G. Danalatos

**Affiliations:** Laboratory of Agronomy and Applied Crop Physiology, Department of Agriculture Crop Production and Rural Environment, University of Thessaly, Fytokou Street, 38446 Volos, Greece

**Keywords:** *Panicum virgatum*, pea, vetch, nitrogen, feedstock, protein content, intercropping

## Abstract

Nowadays, the lack of cattle feed, particularly green fodder, has become a key limiting factor in the agricultural economy. Switchgrass appears to offer a viable solution to the feed shortage. An improved cultivation practice might be needed to boost switchgrass forage production all season long. This study was conducted to quantify the positive effects of introducing different legume crops (vetch and pea), optimally fertilized, on the production and quality of mixed harvested switchgrass–legumes hay in late spring (May) and switchgrass hay harvested once more in early fall (September). The studied intercropping systems, independently of the legume species used, increased forage productivity (almost threefold), reaching 7.5 t ha^−1^ and quality characteristics, with protein content almost rising threefold, reaching 12.5%. The aforementioned practice can assist the perennial crop (switchgrass) in providing a high hay production during the early fall harvest, even without fertilization. The overall annual economic benefit for the farmers may be increased by 90–720 € per ha, depending on the prevailing weather conditions. Overall, it may be concluded that the suggested cropping system produces a significantly higher yield of cattle feed compared to traditional monocultures, improving the agricultural economy while reducing the negative effects of modern agriculture on the environment.

## 1. Introduction

*Panicum virgatum* L. (switchgrass) is a multi-purpose perennial crop. It is mostly used for feed and biofuel production [1,2,3,4,5]. It may also be used as a remediation plant [6,7,8] and is ideal for ground cover and wildlife refuge [9].

As a C_4_ crop, switchgrass is characterized by high biomass production potential. Due to its perennial growth, apart from its high productivity, it also performs high adaptability and improved soil conservation. In numerous studies, switchgrass is characterized as an ideal crop for bioenergy production [10,11,12]. Furthermore, switchgrass can create considerable biomass yields of good quality under low inputs [3] during dry-hot summer days when cool-season grasses are unproductive, providing an important supply of fodder during the summer slump [13,14].

In many countries, a scarcity of livestock feed, especially fodder, has become a major limiting factor in the long-term management of the livestock business [15]. Switchgrass appears to be the greatest option for solving fodder scarcity, given that it can thrive on marginal lands with reduced inputs and maximum outputs [16]. Crop yields in marginal lands, on the other hand, are likely to be lower. As a result, soil fertility control is essential for promoting quick switchgrass growing and preserving system viability [17,18]. Under the correct cultivation scenario and with a little fertilizer reinforcement, switchgrass could supply enough animal feed throughout the growing season.

Nitrogen (N) is an important nutrient for switchgrass production [19,20]; the amount of N required is largely determined by the productivity and yield potential of cultivars under certain conditions and harvest management strategies. The harvest management method has a considerable impact on feedstock quality as well as overall yield, especially in the case of perennial biomass grasses. Crude protein (CP), crude fiber, neutral detergent fiber (NDF), and dry matter digestibility (DMD) are essential markers of the nutritional status of warm-season grasses. Their values decrease as the crop tends to mature [21,22]. In warm-season grasses, the amount and time of N application have a significant and discernible effect on CP content, although NDF, acid detergent fiber (ADF), and ash content are extremely variable [23]. Warm-season grasses are frequently referred to as southern grasses since they thrive in hot summer climates and lack the cool-season grasses’ winter hardiness. Warm-season grasses grow rapidly from mid to late spring through summer and into early October, depending on the region. During winter, they typically turn brown and enter dormancy.

The majority of crop fertilization research has concentrated on nitrogen fertilization, which is typically the most limiting element in nutrition [1,24,25,26]. The addition of nitrogen fertilizer to switchgrass has been shown to increase aboveground biomass yield [4,18,27]. However, excessive N fertilizer use can have negative environmental consequences as well as increase production costs [28]. In addition, soil phosphorus and potassium levels should be maintained, and phosphorus (10 ppm) and potassium (90 ppm) shortages should be corrected with the appropriate application amounts.

In a number of studies, innovative cultivation systems, such as mixed intercropping, have been investigated for reducing nitrogen fertilization rates and producing animal feed with higher protein. Mixed intercropping concerns the use of two or more plant species in the field without any specific layout. Relevant studies with *Amaranthus cruentus—Vigna unguiculata* L. (Amaranth-cowpea) [29,30], cowpea-corn [31], etc., have proven the advantage of intercropping versus monoculture on crop productivity due to the synergetic effect of the growing crops.

It is feasible to intercrop annuals with annuals, annuals with perennials, and perennials with perennials without difficulty in the establishment and cultivation practices [32]. Several scientific works have shown the superiority of *Poaceae* (cereal) and *Fabaceae* (legume) families intercropping systems versus the traditional monocultures. However, little is known about the forage production and quality of switchgrass when it is introduced in a mixed intercropping system. Therefore, the goals of this study were to investigate the effect of introducing different legume crops, such as *Vicia sativa* L. (vetch) and *Pisum sativum* L. (pea), in mixed intercropping systems with *Panicum virgatum* L. (switchgrass) on the production and quality of the harvested hay, under different NPK application levels. It should be noted that the harvested hay in late spring (May) will comprise a mixed switchgrass–legume biomass in late spring (May) and only switchgrass biomass in early fall (September).

## 2. Results

### 2.1. Climatic Data

The study area is characterized by a typical Mediterranean climate, with cool, humid winters and hot, dry summers. Average air temperature during the summer growing periods fluctuated around 25.8 °C, slightly lower than the climatic value (26.2 °C). During the sowing of legumes in 2019, lower-than-average air temperatures prevailed, resulting in a delayed vegetation, especially for the more thermophile plants (Figure 1). After the end of January 2019, normal temperatures prevail again despite minor variations. On the other hand, the temperatures that persisted during the sowing of legumes in 2020 were higher than average (Figure 1). As a result, germination occurred more quickly, allowing the legumes to reach a higher stage of growth and deal well with the low temperatures of the winter months.

Rainfall was about 60 mm lower than average (135 vs. 192 mm) in the period from the sowing of legumes until April (viz. flowering period) in 2019. About 50% of this deficiency was recorded in March (2019), a critical period for plant growth after the winter dormancy. This resulted in the enlargement of the biological cycle and the growth rate reduction of the plants. Fortunately, much of the rain that fell in the first few days of April helped plants to recover and boosted their growth. In the following year (2020), rainfall fluctuated at greater than the climatic levels throughout the growing period, providing rather favorable circumstances for legume growth and productivity.

Average precipitation was recorded during the summer of 2019 (46 mm, Figure 1). However, showers occurring in the second 10-day period of July (2019) brought 28 mm of rain and a noticeable drop in air temperature.

On the other hand, there was a paucity of rainfall throughout the summer months of 2020 (total rainfall of only 9 mm; Figure 1), indicating that a much lower yield might be expected in the second harvest of switchgrass.

In addition, it appears that the average precipitation in the study area was distributed more smoothly in the summer months, while during the experimental year (2019), 60% of the precipitation occurred in the second ten days of July (Figure 1). The low precipitation that occurred in 2020 marked the growing year as very dry and adverse.

### 2.2. Harvested Yield and Quality Characteristics

#### 2.2.1. First Harvest (May)

Dry biomass obtained during spring harvest is shown in Table 1 for both experimentation years. In comparison to switchgrass monoculture, in-cropping treatments with legumes provide statistically significant biomass differences. In particular, the switchgrass monoculture system’s produced dry biomass remained at 33% of the dry yield attained in the intercropping system in both growth years (Table 1). According to the interactions of the tested factors, fertilization improved only the vetch treatments (not the pea treatments). Table 1 demonstrates that the intercropping system of pea with switchgrass under medium fertilization resulted in a greater yield of almost 10 tons per hectare (40-40-40 kg ha^−1^). In addition, the lower air temperatures recorded during the sowing of legumes in 2019, as opposed to the higher air temperatures recorded during the sowing of legumes in 2020, help to explain the improved growth and increased yield obtained in the intercropping system. Additionally, the achieved differences in yield (more than the twofold yield; Table 1) between the study years are entirely explained by the combination of the current air temperatures with the deficit precipitation till April 2019 (a significant element for the growth of winter plants like legumes). Particularly, the dry yield obtained in 2019′s harsh winter, with its decreased precipitation and low temperature during the early stages of legume growth, was approximately half of that in 2020.

Comparing switchgrass with and without legumes, it is clear that the legume–switchgrass mixtures are statistically significant in terms of hay quality features (Table 2). Unlike the interactions between the experiment’s components, the qualitative characteristics of the biomass were not statistically affected by the various fertilization levels. Raised fertilization led to higher protein levels in the vetch–switchgrass treatment; however, this difference was not statistically significant.

Switchgrass–legumes hay, for instance, has a higher protein content (about 12%) than switchgrass hay (about 5%) produced in a monoculture. Specifically, the protein content of switchgrass in-cropping system with legumes is significantly greater (>7%) than that of switchgrass hay (Table 2), making this farming system more competitive since raising the protein content of a feed is a key aspect.

The ash for the co-cultivation system also seems to have risen only in 2019 (from 5% up to 7%), while in 2020, it remained at the same levels. Legumes are low in crude fibers but rich in NDF and ADF. ADF, or the forage’s cellulose and lignin-based cell walls, is constant across all systems and stays at the same levels (26–28% in both cultivating years). The forage digestion capacity of animals is indicated by the aforementioned values of the measured qualitative characteristics.

Finally, it would be interesting to evaluate how promising the studied intercropping systems of switchgrass with vetch or pea, performing by 2–5 t ha^−1^ greater hay yield, depending on the prevailing weather conditions. It should be noted, at this point, that the cost of establishing vetch and pea into a switchgrass crop field is around 338 and 313 € ha^−1^, respectively, considering current market prices. Particularly, the sowing cost (man-and-machine) is approximately 100 € ha^−1^, whereas seeding material of vetch and pea (140 kg ha^−1^) may be purchased against 238 € ha^−1^ (1.7 € kg^−1^) and 213 € ha^−1^ (1.52 € kg^−1^), respectively. Assuming that hay of similar quality (e.g., oats hay with protein value of 8.7–13%) may be sold for 210 € t^−1^, a gross margin of 2–5 t * 210 € t^−1^ = 420–1050 € may be attained per ha annually only from the (late) spring harvest. Subtracting the extra cost involved (average 330 € ha^−1^), the farmer‘s profit may be increased by 90–720 € ha^−1^, annually.

#### 2.2.2. Second Harvest (October)

The second harvest of hay concerns only the switchgrass biomass. In 2019, a superiority of pea, as well as vetch treatment over switchgrass monoculture (although not statistically evident), over switchgrass and legume mixed cultivation treatments (Table 3) was found. In particular, the switchgrass hay output increased by around 6–7 tons per ha when legumes were involved as intercrop, contrary to about 4.5 tons of switchgrass per hectare when the (traditional) monoculture system was used (Table 3). On the other hand, considerable variations among the analyzed agricultural techniques were found in 2020. In the treatments where pea was grown as intercrop throughout the winter, switchgrass performed a significantly higher dry yield (5.4 t ha^−1^; Table 3). It is also evident that the switchgrass hay output of the treatments, where an in-cropping method was not used throughout the winter, remained at low levels of roughly 2.85 t ha^−1^. The decreased precipitation that was observed in the summer of 2020 can explain the high year-to-year fluctuation (Figure 1).

Concerning N-fertilization after the first harvest of switchgrass, dressing with 100 kg of N per ha boosted switchgrass biomass, but only in 2019 independent of the cropping system, obviously as a result of the favorable weather prevailing in the period between the first and second cut (harvest).

No significant differences were found in any factor or combination of factors in terms of protein and ash content (Table 4). Depending on the growing year, the average values of protein and ash content were approximately 5–6% and 4–7%, respectively. The percentages of NDF, ADF, and crude fiber increased significantly by 2% after the supply of 100 kg of nitrogen fertilizer. The average protein content of switchgrass hay, which ranges from 5 to 6.5%, may be seen to be independent of prior cultivation methods and fertilization treatments, despite the noted slight superiority. Additionally, the recorded ash level ranged from 4% (2019) to 8% (2020). NDF and ADF ranged between 50–60% and 30–40%, respectively (Table 4).

## 3. Discussion

The dry yield obtained in the adverse year 2019 (see Section 2.2.1) was similar to the yields reported in the literature concerning mixed cropping with vetch and oat (*Avena sativa* L.) and vetch with barley (*Hordeum vulgare* L.) [33]. In another study [34], biomass production was evaluated among intercropping annual cereals and legumes, intercropping legumes, and monocultures of grains and legumes. This study has shown that intercropping between cereals (oats, spring barley) and legumes (vetch and lupine) resulted in higher biomass production compared to the production of either crop grown in monoculture. In particular, vetch in mixed culture with oats produced more biomass than the aforementioned species grown separately. Furthermore, the different fertilization levels did not significantly affect the yield, and this may be attributed to the switchgrass's low input requirements on the one hand and on the other by the fact that nitrogen-fixing legumes might have supplied a portion of the N diet for switchgrass production. Similar experiments with legume–switchgrass mixed stands resulted in yields that exceeded those of monoculture [35], in line with the results of the present study. According to previous experiments, cultivating a variety of plants promotes resource conservation, while the synergies between productivity and biodiversity are also strengthened [36].

Switchgrass–legumes mixed cropping produced hay with protein content increased almost threefold than switchgrass monoculture, which is in line with the literature reporting that vetch–cereal mixed cropping is highly productive in protein-rich aerial biomass [37]. The current study’s findings demonstrate that a zero-fertilization switchgrass monoculture system generates animal feed with a protein content of about 5%, which is in perfect agreement with a previous survey conducted in the same location, despite the crop being in its second and third growing year [38].

As already noted, the forage’s cellulose- and lignin-based cell walls are known as ADF. Legumes have modest levels of crude fibers but high levels of NDF and ADF [39]. The ability of animals to digest the forage is characterized by the values of the aforementioned traits. ADF data are used to produce many of the calculated numbers on the forage reports because forage digestibility often decreases as ADF increases. It is reported [40] that most of the cereals’ hay has an ADF in a range of 36–42%; however, in this study, the harvested hay remained at levels below 30% even in the case of switchgrass as a monocrop, enhancing a good quality straw for animal feed. The NDF value represents the complete cell wall, which includes both the ADF and hemicellulose fractions. In the rumen, cellulose and hemicellulose are partially digested, whereas lignin is an indigestible fiber [41,42]. In this study, NDF remained at lower levels, too, compared to hay produced by other crops [43,44]. Furthermore, in the case of vetch as a monocrop, Karsli et al. [45] and Larbi et al. [46] reported that the contents of protein, NDF, and ADF in vetch hay were similar to the values found in this study.

Finally, the current study’s qualitative features (protein, ADF, and NDF) were superior to forage nutritive values reported by other studies in the case of the produced mixed forages (switchgrass with legumes) [47,48,49,50].

It is clear that upon second harvest (following a mixed cropping with legumes), a remarkable switchgrass hay yield of roughly 5 tons per ha may be produced, even without any N-fertilization. Such hay yields reach or even exceed the hay outputs recorded from many other cultivated cereals (e.g., oats [*Avena sativa* L.], rice [*Oryza sativa* L.], wheat [*Triticum* sp. L.], and triticale [*Triticosecale*, Wittmack] [51]).

Independent of cultivation practice, the protein content measured in switchgrass forage (5–6%) is substantially greater than the values reported for rice hay [52], barley (*Hordeum vulgare* L., 4.3%) [53], and wheat (3.5%) [54], but lacks behind the protein content reported for oats hay by 1–1.8% [55]. Nevertheless, the digestibility of switchgrass hay, as measured by ADF and NDF values, appears to be greater than that of oats [55].

Furthermore, the ash content values measured in this study were lower than those reported by Mahmood et al. [56] for other cereal crops. Filya [57] stated that there is a decline in ash content following plant maturation so any comparison of ash contents should be made, if possible, at the same growing stage. Finally, the NDF and ADF ranges of switchgrass hay found here match perfectly with the values documented in the literature for crops belonging to the same family (*Poaceae*), such as sorghum (*Sorghum bicolor*, L.) with 48.3–55.4% and 21.7–37.0%, for NDF and ADF, respectively [58]. When only switchgrass hay was evaluated, these results verified the outcomes of the current investigation.

## 4. Materials and Methods

### 4.1. Experimental Site

A two year field experiment was established in the Experimental Farm of the University of Thessaly, situated in Velestino (Magnesia) area, having a typical Mediterranean climate such as that prevailing in (east) Thessaly, Central Greece (coordinates 39°02’ N, 22°45’ E). The experiment was carried out during the growing periods of 2018–2019 and 2019–2020, when switchgrass was in its 9th and 10th growing year, respectively.

An automated meteorological station was placed in the vicinity of the experimental field to collect complete weather data.

### 4.2. Soil Characteristics

The experimental site contains a particularly fertile (organic matter of 2.91% at a depth of 0–30 cm and 1.86% at 30–60 cm), calcareous clayey soil with an alkaline reaction (Table 5) classified, as *Calcixerollic Xerochrept*, according to USDA Soil Taxonomy [59]).

### 4.3. Experimental Design

A factorial split-plot design was used in three replicates (blocks) and twenty-seven plots per replication. Main factor comprised three different cultivation systems (S_1_: switchgrass monoculture, S_2_: switchgrass–vetch intercropping, S_3_: switchgrass–pea intercropping). Sub-factor comprised three different fertilization levels (F_1_: 0, F_2_: 50 kg N and 40 kg PK per ha, F_3_: 100 kg N and 80 kg PK per ha). Fertilization was applied after the first harvest of the crop using an NPK (15-15-15) mixed fertilizer and urea (40-0-0). Each experimental plot is sized 66 m^2^ (11 m × 6 m).

The levels of fertilization dressings were decided based on NPK concentration of switchgrass tissue found in previous studies in the same area (e.g., 0.8% N, 0.08% P, and 1.3% K, and potassium, respectively) [38]. Considering a deep rooting system obtained after 9 and 10 years of growth, the applied levels PK are expected to be rather supporting to the intercropped legume crop. Finally, it is reported that switchgrass tissue N concentration is not affected by N rates [60].

### 4.4. Crop Management Practices

The sowing of vetch and pea took place on 21 November 2018 and 10 December 2019 using a modern no-tillage seeding cereal machine, while switchgrass cultivation was in its 9th and 10th growing year (switchgrass row space: 12.5 cm), respectively. The “Olympus” pea variety [61] and the “Alexandros” vetch variety [62], with a quantity of 140 kg ha^−1^, were used in both plants (Figure 2). Due to the fact that switchgrass was in its ninth and tenth growing years and was more competitive, no pesticides were used; however, all plots were hand-weeded, and no pesticides were used during the establishment year (2009–2010).

Legumes and switchgrass were harvested at the end of May (29 May 2019 and 25 May 2020; Figure 3), where legumes were at flowering stage and switchgrass at the development of the 5th–6th leaf. Nitrogen fertilization was then performed according to the experimental design, as well as irrigations for the production of switchgrass biomass. Irrigation was carried out through summer period by sprinkling in four equal supplies of 50 mm each as supplementary irrigation. The harvest took place on 25 October 2019 and 13 November 2020, where switchgrass was again at the development of the 5th–6th leaf.

The plants were cut 8–10 cm above ground using four frames with dimensions 50 cm × 50 cm each, and a total area of 1 m^2^ was harvested. To prevent any border effect, wire frames were positioned at random locations within the plots. Two persons moved the biomass by hand until the sides of the frame could be placed on the ground, and the contained plant material could be cut. The samples were weighed in the field, and then a sub-sample (1/2 of the total sample weight) was taken for further laboratory measurements and air drying at a temperature of ≤60 °C.

### 4.5. Yield Quality Characteristics Measures

To be prepared for the DA 7250 NIR analyzer (Perten Instruments, Hägersten, Sweden) measurements of the quality characteristics, the dry samples were chopped and ground in the laboratory. This analyzer uses a near-infrared (NIR) spectroscopic approach and has the required curves stored in its software. The samples’ ash, protein, neutral detergent fiber (NDF), acid detergent fiber (ADF), calcium content, and phosphorus content were all measured. NDF and ADF offer suggestions for plant quality parameters based on crop age and growth stage. Using NDF and ADF measurements, the amount of lignin, cellulose, hemicelluloses, and insoluble minerals in the feed is determined.

### 4.6. Statistical Analysis

Finally, the statistical program GenStat (7th Edition) was used to perform an analysis of variance (ANOVA) within sample timings for all measured and calculated variables. The LSD_.05_ was used as the test criterion for determining differences between means of the main and/or interaction effects [63].

## 5. Conclusions

Intercropping systems and particularly a legume–switchgrass sod-seeding cropping system was proved to be a very effective strategy for increasing forage productivity as well as forage quality. Actually, such a cultivation system might triple the quantity of the hay produced in the first cut (spring harvest) while also tripling the protein concentration. Furthermore, the aforesaid system favors high productivity and final hay yields also in the second cut (autumn harvest) even under no N-fertilization.

Additionally, depending on the prevailing weather conditions, the increased productivity and obtained yields ensure a greater farmers’ income as compared to the existing traditional monocultures. Farmers will be able to apply less fertilization and herbicides inputs reducing cultivation costs even further while enhancing biodiversity and environmental protection. This is of great importance for agricultural development at the local and regional levels.

Overall, it may be concluded that the suggested cropping system involving mixed cultures of perennial cereals and annual winter legumes produces a significantly higher yield of cattle feed, improving the economic standing of the farmers while reducing the negative environmental effects of modern high-input agriculture.

## Figures and Tables

**Figure 1 plants-11-02970-f001:**
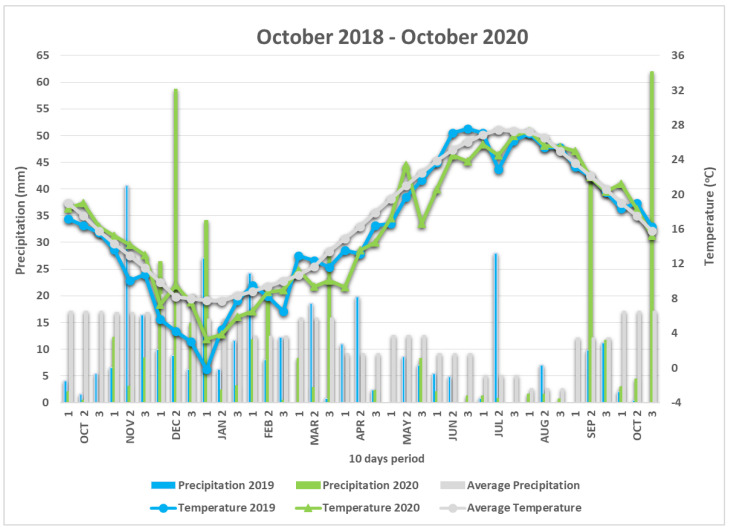
Average ten days air temperature-precipitation of the study site in 2019–2020 and average air temperature-precipitation of the last 30 years.

**Figure 2 plants-11-02970-f002:**
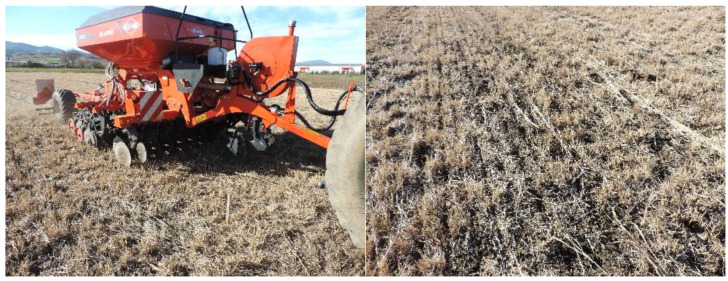
Pea and Vetch sow using no-till machine in an existing switchgrass cultivation.

**Figure 3 plants-11-02970-f003:**
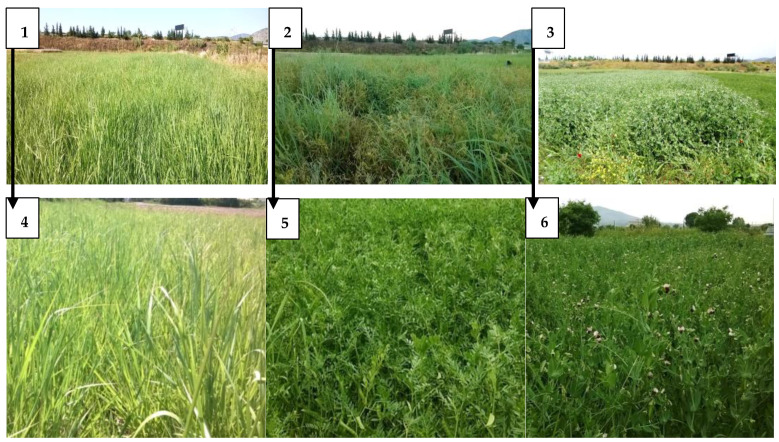
S_1_: switchgrass as monoculture (**1**,**4**), S_2_: switchgrass with vetch (**2**,**5**), S_3_: switchgrass with pea (**3**,**6**). In all cases, switchgrass is in the growing stage of 5–6 leaves.

**Table 1 plants-11-02970-t001:** Dry biomass yield of switchgrass in a monoculture and co-culture system of vetch or pea during 2019 and 2020.

	Variables	Dry Weight (kg ha^−1^)	
Treatments		2019	2020
Cover Crops	Switchgrass (S)	1180	2560
	Vetch (V)	3770	7040
	Pea (P)	4130	7690
*LSD* _.05_		*770*	*2909*
N-P-K (kg ha^−1^)	0	2830	5500
	40-40-40	2990	6270
	80-80-80	3260	5520
*LSD* _.05_		*ns*	*ns*
Interaction	S0	1170	2280
	S40-40-40	1290	2480
	S80-80-80	1080	2920
	V0	3080	7080
	V40-40-40	3550	6830
	V80-80-80	4680	7210
	P0	4240	7140
	P40-40-40	4140	9490
	P80-80-80	4020	6430
*LSD* _.05_		*ns*	*ns*
*CV (%)*		*24.0*	*22.3*

*LSD*
_.05_
*: Least Significant Difference (5%), CV: Coefficient of Variance, ns: non-significant.*

**Table 2 plants-11-02970-t002:** Qualitative characteristics of switchgrass biomass in monoculture and co-culture of vetch or pea.

	Variables	Protein		Ash		NDF		ADF		Crude Fiber	
Treatments		2019	2020	2019	2020	2019	2020	2019	2020	2019	2020
Cover Crops	Switchgrass (S)	5.30	5.17	5.19	4.68	48.10	47.65	27.99	28.22	32.01	32,12
	Vetch (V)	12.69	12.21	7.33	4.85	47.21	48.72	26.64	28.85	28.13	34.53
	Pea (P)	11.86	13.73	6.60	4.32	44.97	48.02	25.19	28.43	29.30	35.86
*LSD* _.05_		*1.523*	*2.541*	*0.652*	*ns*	*2.321*	*ns*	*1.705*	*ns*	*2.452*	*2.423*
N-P-K (kg ha^−1^)	0	9.59	9.89	6.38	4.5	47.30	48.03	27.03	28.43	30.13	34,71
	40-40-40	9.89	10.82	6.27	4.66	46.45	47.01	26.40	28.23	29.89	33,91
	80-80-80	10.36	10.40	6.47	4.69	46.52	48.55	26.39	28.85	29.42	34,28
*LSD* _.05_		*ns*	*ns*	*Ns*	*ns*	*ns*	*ns*	*ns*	*ns*	*ns*	*ns*
Interaction	S0	5.67	5.14	5.35	4.66	48.53	48.14	28.26	28.59	32.12	33.26
	S40-40-40	5.45	5.34	5.23	4.69	48.30	46.72	28.12	27.5	32.43	30.87
	S80-80-80	4.77	5.02	4.99	4.7	47.48	48.1	27.59	28.58	31.47	32.22
	V0	11.89	10.95	7.27	4.79	47.93	49.17	27.18	29.2	29.07	34.99
	V40-40-40	12.45	12.87	7.00	4.85	45.54	48.36	25.50	28.52	27.17	35.21
	V80-80-80	13.72	12.80	7.70	4.91	48.17	48.62	27.24	28.84	28.14	34.58
	P0	11.22	13.57	6.52	4.04	45.45	46.78	25.64	27.52	29.20	35.89
	P40-40-40	11.76	14.23	6.58	4.44	45.52	48.35	25.58	28.66	30.05	35.65
	P80-80-80	12.60	13.39	6.71	4.48	43.92	48.94	24.34	29.12	28.64	36.03
*LSD* _.05_		*ns*	*ns*	*Ns*	*ns*	*ns*	*ns*	*ns*	*ns*	*ns*	*ns*
*CV (%)*		*11.8*	*12.8*	*10.2*	*5.1*	*4.2*	*2.7*	*5.2*	*3.5*	*3.8*	*4.9*

*NDF: Neutral Detergent Fiber, ADF: Acid Detergent Fiber, LSD*
_.05_
*: Least Significant Difference (5%), CV: Coefficient of Variance, ns: non-significant.*

**Table 3 plants-11-02970-t003:** Dry yield at the second harvest (October cutting) in switchgrass cultivation derived from a monoculture and mixed culture with vetch or pea under different N application rates during the cultivation years 2019 and 2020.

	Variables	Dry Weight kg ha^−1^	
Treatments		2019	2020
Cover Crops	Switchgrass (S)	4380	2850
	Vetch (V)	6080	4340
	Pea (P)	7210	5400
*LSD* _.05_		*ns*	*677*
Nitrogen (kg ha^−1^)	0	3980	3880
	50	5530	4350
	100	8160	4360
*LSD* _.05_		*1769*	*ns*
Interaction	S0	3390	2590
	S50	4900	2500
	S100	4850	3470
	V0	4240	4280
	V50	5770	3870
	V100	8210	4850
	P0	4310	4760
	P50	5920	6670
	P100	11,410	4770
*LSD* _.05_		*ns*	*ns*
*CV (%)*		*29.2*	*32.7*

*LSD*
_.05_
*: Least Significant Difference (5%), CV: Coefficient of Variance, ns: non-significant.*

**Table 4 plants-11-02970-t004:** Qualitative characteristics of the harvested biomass at the second cut (October) in switchgrass cultivation derived from a monoculture and mixed crop with vetch or pea under the regime of different (nitrogen) fertilization levels during the cultivation years 2019 and 2020.

	Variables	Protein		Ash		NDF		ADF		Crude Fiber	
Treatments		2019	2020	2019	2020	2019	2020	2019	2020	2019	2020
Cover Crops	Switchgrass (S)	5.03	5.50	4.23	8.01	50.47	60.06	29.76	38.88	34.27	28.44
	Vetch (V)	5.15	6.47	4.22	7.64	51.10	59.60	30.29	39.04	35.38	29.37
	Pea (P)	5.04	5.30	4.19	7.40	51.55	60.05	30.59	39.55	35.69	30.04
*LSD* _.05_		*ns*	*ns*	*ns*	*0.460*	*ns*	*ns*	*ns*	*ns*	*ns*	*1.34*
Nitrogen (kg ha^−1^)	0	5.05	5.43	4.14	7.54	49.68	59.54	29.17	39.27	34.15	29.41
	50	5.09	6.02	4.24	8.00	50.97	60.41	30.18	38.96	35.05	29.01
	100	5.08	5.82	4.27	7.51	52.47	59.76	31.30	39.77	36.13	29.44
*LSD* _.05_		*ns*	*ns*	*ns*	*0.486*	*1.385*	*ns*	*1.769*	*ns*	*1.116*	*1.579*
Interaction	S0	5.04	5.17	4.26	7.60	49.74	58.89	29.17	38.58	33.41	29.11
	S50	5.14	6.25	4.26	8.79	49.97	60.68	29.39	38.29	33.82	28.29
	S100	4.90	5.09	4.18	7.65	51.70	60.61	30.73	39.77	35.57	27.93
	V0	5.10	5.86	4.23	7.46	50.29	59.77	29.67	39.34	34.63	29.81
	V50	5.27	7.62	4.31	7.91	50.97	59.05	30.21	38.30	35.02	28.61
	V100	5.07	5.92	4.13	7.66	52.03	59.98	31.01	39.49	36.49	29.68
	P0	5.01	5.28	3.92	7.57	49.00	59.97	28.67	39.89	34.40	29.30
	P50	4.85	4.18	4.16	7.41	51.97	61.50	30.95	40.29	36.32	30.12
	P100	5.26	6.44	4.50	7.22	53.69	58.69	32.16	38.46	36.34	30.70
*LSD* _.05_		*ns*	*ns*	*ns*	*ns*	*ns*	*ns*	*ns*	*ns*	*ns*	*ns*
*CV (%)*		*7.9*	*27.3*	*6.0*	*6.2*	*2.6*	*3.3*	*3.3*	*3.4*	*3.1*	*5.2*

*NDF: Neutral Detergent Fiber, ADF: Acid Detergent Fiber, LSD*
_.05_
*: Least Significant Difference (5%), CV: Coefficient of Variance, ns: non-significant.*

**Table 5 plants-11-02970-t005:** Soil properties of surface (0–30 cm) and sub-surface horizons (30–60 cm).

	pH	Composition			Organic Matter %	Bulk Density	Cation Exchange Capacity cmol kg^−1^	C/N
		Sand %	Silt %	Clay %				
0–30 cm	7.6	26.8	31.3	41.9	2.91	1.27	8	8.8
30–60 cm	7.9	25.9	30.9	43.1	1.86	1.27	4	8.9

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
