# Peer review of "Could a Legume–Switchgrass Sod-Seeding System Increase Forage Productivity?"

_plants, 2022, doi:10.3390/plants11212970_

Round 1

Reviewer 1 Report

Dear colleagues,

This review is concerning a research work entitled “Could a legume-switchgrass sodseeding system increase forage productivity, by Kyriakos D. Giannoulis, Dimitrios Bartzialis, Elpiniki Skoufogianni, Ippolitos Gintsioudis and Nicholaos G.  Danalatos.  As detailed data and experiments increasing the knowledge of plants, I recommend it for an international audience in this journal, however several points have to be precised and a major revision is requested.

Please notice that in order to bring a broad audience to this article and to this journal, for specialists and non-specialists, the six major points of my comments (at the beginning) are very important (mandatory…) for a suitable value of the article. Minor points are also enhanced at the end of this review.

I deeply hope to see this good article published soon,

The six major points are:

 1-     - Although I am not a native english speaker, this manuscript has to be re-written for proper british or american english (please check with the journal for this choice); many sentences look like word-for-word translation (some examples are given in the minor points) and it is not acceptable for an international journal devoted to proper english language.

22-     As I am involved in plant taxonomy I am very sensible to correct taxa names, which should be inserted at least the first time they appear in the text. So from the introduction, insert latin names in italics (and author(s)) of all plants cited (species and variety (for these latters Olympus and Alexandros, if the author(s) does(do) not exist put the first reference where they appeared in the scientific literature)); check them one by one all through the text, even if cited only once like  “oats, rice,  wheat… and others in the discussion.  Insert also the families as for instance Poaceae and Fabaceae have a completely different biology. Use international Plant Names Index (IPNI) https://www.ipni.org/), or equivalent.

33-     In order to be more attractive especially for non-specialists, as it is a journal devoted to plants a figure with photos of the plants involved (and of (some of ) the experiments if available) is necessary.

44-     In the results, in order to be understood more rapidly use more values to describe the tables and etc, even if they are repeated several times in the text. Moreover restrict the results part to the only the description of values and etc, and remove all sentences which belong actually to the discussion part (e.g. "making this cropping system more competitive in high feed output.", "These numbers refer to an animal's ability to digest the forage.", "The difference that was found between the cultivating years can be explained...", "100 kilograms of nitrogen increased fresh biomass". Check these in all parts of the results.

55-     In the conclusion (or end of discussion), it would be great, as an added value to these interesting data, in order to provide to farmers or others a practical strategy in their fields, to insert a synthetized scheme with precise propositions of (simplified) values (or percentages or whatever) of (combined between the different plants) sodseedings for instance, also in terms of quality of soil or environment features, inducing a (putative) increase of production thanks to your research. Add this point in the abstract as it may be very attractive for the readers.

66-     References already taken in account by the authors are of real interest, however checking briefly in the word of science WOS and scilit (from mdpi) with the key-words of the abstract, other references (among them very recent ones as your most recent one is from 2019…) appear and references should be once more selected and used (if relevant…) in order to provide a larger view of this interesting research. Among these are the followings:

[1-32]

1.         Alexopoulou, E.; Papatheohari, Y.; Christou, M. Switchgrass a Perennial Grass with Sustainable Biomass Yields that Can Be Cultivated in Most European Climatic Areas.

2.         Alexopoulou, E.; Papatheohari, Y.; Christou, M.; Soldatos, Y. Switchgrass - A Valuable Perennial Grass for Europe for Both Marginal and Agricultural Lands.

3.         Alexopoulou, E.; Sharma, N.; Papatheohari, Y.; Christou, M.; Piscioneri, I.; Panoutsou, C.; Pignatelli, V. Biomass yields for upland and lowland switchgrass varieties grown in the Mediterranean region. Biomass and Bioenergy 2008, 32.

4.         Alexopoulou, E.; Zanetti, F.; Papazoglou, E.G.; Christou, M.; Papatheohari, Y.; Tsiotas, K.; Papamichael, I. Long-term studies on switchgrass grown on a marginal area in Greece under different varieties and nitrogen fertilization rates. Ind Crop Prod 2017, 107.

5.         Alexopoulou, E.; Zanetti, F.; Papazoglou, E.G.; Iordanoglou, K.; Monti, A. Long-Term Productivity of Thirteen Lowland and Upland Switchgrass Ecotypes in the Mediterranean Region. Agronomy 2020, 10.

6.         Alexopoulou, E.; Zanetti, F.; Scordia, D.; Lizarazu, W.Z.; Christou, M.; Testa, G.; Cosentino, S.; Monti, A. Long-Term Yields of Switchgrass, Giant Reed, and Miscanthus in the Mediterranean Basin. BioEnergy Research 2015, 8.

7.         Ashworth, A.; Allen, F.L.; Keyser, P.D.; Tyler, D.D.; Saxton, A.; Taylor, A.M. Switchgrass yield and stand dynamics from legume intercropping based on seeding rate and harvest management. Journal of Soil and Water Conservation 2015, 70.

8.         Ashworth, A.J.; Allen, F.L.; Goddard, K.; Warwick, K.S.; Yeaman, D.; Pote, D.H. Switchgrass Compositional Variations Arising from Spatial Distribution and Legume Intercropping. Communications in Soil Science and Plant Analysis 2017, 48.

9.         Ashworth, A.J.; Allen, F.L.; Warwick, K.S.; Keyser, P.D.; Bates, G.E.; Tyler, D.D.; Lambdin, P.L.; Pote, D.H. N2 Fixation of Common and Hairy Vetches when Intercropped into Switchgrass. Agronomy 2017, 7.

10.       Ashworth, A.J.; Keyser, P.D.; Allen, F.L.; Tyler, D.D.; Taylor, A.M.; West, C.P. Displacing Inorganic Nitrogen in Lignocellulosic Feedstock Production Systems. Agron J 2016, 108.

11.       Ashworth, A.J.; Taylor, A.M.; Reed, D.L.; Allen, F.L.; Keyser, P.D.; Tyler, D.D. Environmental impact assessment of regional switchgrass feedstock production comparing nitrogen input scenarios and legume-intercropping systems. J Clean Prod 2015, 87.

12.       Ashworth, A.J.; Weiss, S.A.; Keyser, P.D.; Allen, F.L.; Tyler, D.D.; Taylor, A.; Beamer, K.P.; West, C.P.; Pote, D.H. Switchgrass composition and yield response to alternative soil amendments under intensified heat and drought conditions. Agriculture, Ecosystems & Environment 2016, 233.

13.       Ashworth, A.J.; West, C.P.; Allen, F.L.; Keyser, P.D.; Weiss, S.A.; Tyler, D.D.; Taylor, A.M.; Warwick, K.L.; Beamer, K.P. Biologically Fixed Nitrogen in Legume Intercropped Systems: Comparison of Nitrogen;Difference and Nitrogen Enrichment Techniques. Agron J 2015, 107.

14.       Cha, G.; Meinhardt, K.A.; Orellana, L.H.; Hatt, J.K.; Pannu, M.W.; Stahl, D.A.; Konstantinidis, K.T. The influence of alfalfa;switchgrass intercropping on microbial community structure and function. Environmental Microbiology 2021, 23.

15.       Collins, H.P.; Fay, P.A.; Kimura, E.; Fransen, S.; Himes, A. Intercropping with Switchgrass Improves Net Greenhouse Gas Balance in Hybrid Poplar Plantations on a Sand Soil. Soil Science Society of America Journal 2017, 81.

16.       Collins, H.P.; Kimura, E.; Polley, W.; Fay, P.A.; Fransen, S. Intercropping switchgrass with hybrid poplar increased carbon sequestration on a sand soil. Biomass and Bioenergy 2020, 138.

17.       Kimura, E.; Fransen, S.C.; Collins, H.P.; Stanton, B.J.; Himes, A.; Smith, J.; Guy, S.O.; Johnston, W.J. Effect of intercropping hybrid poplar and switchgrass on biomass yield, forage quality, and land use efficiency for bioenergy production. Biomass and Bioenergy 2018, 111.

18.       Lychnaras, V.; Schneider, U. Multi-farm economic analysis of perennial energy crops in Central Greece, taking into account the CAP reform. Biomass and Bioenergy 2011, 35.

19.       Mantino, A.; Giannini, V.; Tozzini, C.; Bonari, E.; Ragaglini, G. The overseeding of two cool-season legumes (Hedysarum coronarium L. and Trifolium incarnatum L.) on switchgrass (Panicum virgatum L.) mature stands increased biomass productivity. Italian Journal of Agronomy 2020, 15.

20.       Mantino, A.; Ragaglini, G.; Nasso, N.N.O.D.; Tozzini, C.; Taccini, F.; Bonari, E. Alfalfa (Medicago sativa L.) overseeding on mature switchgrass (Panicum virgatum L.) stand: biomass yield and nutritive value after the establishment year. Italian Journal of Agronomy 2016, 11.

21.       Muwamba, A.; Amatya, D.M.; Chescheir, G.M.; Nettles, J.E.; Appelboom, T.; Tollner, E.; Ssegane, H.; Youssef, M.A.; Birgand, F.; Callahan, T. Response of Drainage Water Quality to Fertilizer Applications on a Switchgrass Intercropped Coastal Pine Forest. Water-Sui 2020, 12.

22.       Muwamba, A.; Amatya, D.M.; Ssegane, H.; Chescheir, G.M.; Appelboom, T.; Nettles, J.E.; Tollner, E.W.; Youssef, M.A.; Walega, A.; Birgand, F. Response of Nutrients and Sediment to Hydrologic Variables in Switchgrass Intercropped Pine Forest Ecosystems on Poorly Drained Soil. Water, Air, & Soil Pollution 2020, 231.

23.       Muwamba, A.; Amatya, D.M.; Ssegane, H.; Chescheir, G.M.; Appelboom, T.; Tollner, E.W.; Nettles, J.E.; Youssef, M.A.; Birgand, F.; Skaggs, R.W.; et al. Effects of Site Preparation for Pine Forest/Switchgrass Intercropping on Water Quality. Journal of Environmental Quality 2015, 44.

24.       Panda, S.S.; Amatya, D.M.; Muwamba, A.; Chescheir, G. Estimation of evapotranspiration and its parameters for pine, switchgrass, and intercropping with remotely-sensed images based geospatial modeling. Environmental Modelling & Software 2019, 121.

25.       Pannu, M.W.; Meinhardt, K.A.; Bertagnolli, A.; Fransen, S.C.; Stahl, D.A.; Strand, S.E. Nitrous oxide emissions associated with ammonia;oxidizing bacteria abundance in fields of switchgrass with and without intercropped alfalfa. Environmental Microbiology Reports 2019, 11.

26.       Paschalidou, A.; Tsatiris, M. SWOT Analysis of Perennial vs Annual Energy Crops: A Case Study in Greece; 2021.

27.       Scordia, D.; Zanetti, F.; Varga, S.S.; Alexopoulou, E.; Cavallaro, V.; Monti, A.; Copani, V.; Cosentino, S.L. New Insights into the Propagation Methods of Switchgrass, Miscanthus and Giant Reed. BioEnergy Research 2015, 8.

28.       Strickland, M.S.; Leggett, Z.H.; Sucre, E.B.; Bradford, M.A. Biofuel intercropping effects on soil carbon and microbial activity. Ecological Applications 2015, 25.

29.       Sutradhar, A.K.; Miller, E.C.; Arnall, D.B.; Dunn, B.L.; Girma, K.; Raun, W.R. Switchgrass forage yield and biofuel quality with no-tillage interseeded winter legumes in the southern Great Plains. Journal of Plant Nutrition 2017, 40.

30.       Taranenko, A.; Kulyk, M.; Galytska, M.; Taranenko, S. Effect of cultivation technology on switchgrass (Panicum virgatum L.) productivity in marginal lands in Ukraine. Acta Agrobotanica 2019, 72.

31.       Warwick, K.; Allen, F.L.; Keyser, P.D.; Ashworth, A.; Bates, G.E.; Tyler, D.D.; Lambdin, P.L.; Harper, C.A. Biomass and integrated forage/biomass yields of switchgrass as affected by intercropped cool- and warm-season legumes. Journal of Soil and Water Conservation 2016, 71.

32.       Yost, M.A.; Kitchen, N.R.; Sudduth, K.A.; Thompson, A.L.; Allphin, E. Topsoil Thickness Influences Nitrogen Management of Switchgrass. BioEnergy Research 2017, 10.

Minor points are:

1 in the introduction, in the paragraph "Nitrogen (N) is...", precise  which plants are concerned by the "decrease of maturity"; do the same for the next sentence "on warm-season grasses..."; 

2 in 2.1, reword "of the Mediterranean.";

3 in 2.1, reword "to the study area's normal air temperature.";

4 in 2.1, reword "greater above" and put "higher above"?;

5 in 2.1, reword “growth was seen to be slowed.”;

6 for figure 1, put colours to distinguish rapidly the different columns and the curves;

7 for table 1, insert a caption explaining all words and/or symbols-abbreviations of the column "treatments" and "variables", especially “LSD” and “ns” which are important for the discussion;

8 do the same for table 2, also for NDF and ADF;

9 do the same for table 3;

10 do the same for table 4, also for NDF and ADF;

11 in the discussion first paragraph line 2, precise which "similar environments” you are talking about;

12 in the discussion, in the paragraph "legumes are also…”, precise references for the "ability to digest the forage";

13 in the same paragraph, precise references for "In the rumen, cellulose and hemicellulose...";

14 in the discussion last paragraph, why do you separate sorghum from other plants reported just before?; moreover precise the plants reported in references 47 and 48 as apparently they refer to various taxa of various families (?), discuss accordingly these details in this paragraph;

15 for table 5, put in a caption the meaning of "CEC";

16 in 4.4, precise how you placed "in random" the frames; moreover, are you sure of “2009-2010” ?; precise also the weight and/or proportion of “sub-sample”;

17 some references do not seem to be homogeneous in their format, like “2004” in bold letters in “McCartney, D.; Okine, E.K.; Baron, V.S.; Depalme, A.J. Alternative fall and winter feeding systems for spring calving beef cows. 403 Can. J. Anim. Sci., 2004, 84, 511–522. Please check all of them again.

Author Response

Title: "Could a legume-switchgrass sodseeding system increase forage productivity?"

Journal: Plants

Dear Editor,

We are thankful for giving us the opportunity to support our work. Following reviewers suggestion, we are sending the revised manuscript, based on our feeling that the work has been modified taking into account the suggestions made by the reviewers. In the revised manuscript, we have carried out important changes.

All the changes made in the revised manuscript are in colored yellow. For your decision and comparison with the first version, we present the changes in juxtaposition with reviewer’s suggestions in the next pages.

Sincerely yours,

Kyriakos Giannoulis

Checklist

  • Although I am not a native english speaker, this manuscript has to be re-written for proper british or american english (please check with the journal for this choice); many sentences look like word-for-word translation (some examples are given in the minor points) and it is not acceptable for an international journal devoted to proper english language.

Answer: English has been checked

  • As I am involved in plant taxonomy I am very sensible to correct taxa names, which should be inserted at least the first time they appear in the text. So from the introduction, insert latin names in italics (and author(s)) of all plants cited (species and variety (for these latters Olympus and Alexandros, if the author(s) does(do) not exist put the first reference where they appeared in the scientific literature)); check them one by one all through the text, even if cited only once like  “oats, rice,  wheat… and others in the discussion.  Insert also the families as for instance Poaceae and Fabaceae have a completely different biology. Use international Plant Names Index(IPNI) https://www.ipni.org/), or equivalent.

Answer: All Latin names have been included according to the comments

  • In order to be more attractive especially for non-specialists, as it is a journal devoted to plants a figure with photos of the plants involved (and of (some of ) the experiments if available) is necessary.

Answer: Photos have been included according to the suggestions.

  • In the results, in order to be understood more rapidly use more values to describe the tables and etc, even if they are repeated several times in the text. Moreover restrict the results part to the only the description of values and etc, and remove all sentences which belong actually to the discussion part (e.g. "making this cropping system more competitive in high feed output.", "These numbers refer to an animal's ability to digest the forage.", "The difference that was found between the cultivating years can be explained...", "100 kilograms of nitrogen increased fresh biomass". Check these in all parts of the results.

Answer: Appropriate values to the results have been included according to the comments.

  • In the conclusion (or end of discussion), it would be great, as an added value to these interesting data, in order to provide to farmers or others a practical strategy in their fields, to insert a synthetized scheme with precise propositions of (simplified) values (or percentages or whatever) of (combined between the different plants) sodseedings for instance, also in terms of quality of soil or environment features, inducing a (putative) increase of production thanks to your research. Add this point in the abstract as it may be very attractive for the readers.

Answer: A briefly economic analysis for the increase of farmers’ income has been added.

  • References already taken in account by the authors are of real interest, however checking briefly in the word of science WOS and scilit (from mdpi) with the key-words of the abstract, other references (among them very recent ones as your most recent one is from 2019…) appear and references should be once more selected and used (if relevant…) in order to provide a larger view of this interesting research.

Answer: New references have been included.

  • In the introduction, in the paragraph "Nitrogen (N) is...", precise  which plants are concerned by the "decrease of maturity"; do the same for the next sentence "on warm-season grasses...";

Answer: Everything has been explained according to the comments

  •  

Answer: All suggested rewords have been included.

  • for figure 1, put colours to distinguish rapidly the different columns and the curves;

Answer: Figure 1 has been colored according to the suggestions.

  • Table 1-4 insert a caption explaining all words and/or symbols-abbreviations of the column "treatments" and "variables", especially “LSD”, “ns” NDF and ADF which are important for the discussion;

Answer: Appropriate captions have been included according to the comments.

  • In the discussion, in the paragraph "legumes are also…”, precise references for the "ability to digest the forage". In the same paragraph, precise references for "In the rumen, cellulose and hemicellulose...";

Answer: References have been used according to the comments.

  • 14 in the discussion last paragraph, why do you separate sorghum from other plants reported just before?; moreover precise the plants reported in references 47 and 48 as apparently they refer to various taxa of various families (?), discuss accordingly these details in this paragraph;

Answer: It is corrected according to the comments.

  • For table 5, put in a caption the meaning of "CEC";

Answer: It is corrected according to the comments.

  • In 4.4, precise how you placed "in random" the frames; moreover, are you sure of “2009-2010” ?; precise also the weight and/or proportion of “sub-sample”;

Answer: Materials and Methods have been changed according to the comments

  • Some references do not seem to be homogeneous in their format, like “2004” in bold letters in “McCartney, D.; Okine, E.K.; Baron, V.S.; Depalme, A.J. Alternative fall and winter feeding systems for spring calving beef cows. 403  J. Anim. Sci., 2004, 84, 511–522. Please check all of them again.

Answer: All references have been checked according to the comments

Reviewer 2 Report

In this paper, research was undertaken to evaluate the effect of different legume crops (vetch and peas) and the optimal fertilization scenario on the production and quality of mixed hay harvested in late spring (May) and switch millet in early autumn (September).

The topic of the undertaken research seems interesting, however, the presented work, in my opinion, contains some shortcomings.

1. The abstract of the work should include results and not just general statements.

2.The introduction needs to be revised, please provide relevant information to the topic of the work.

3. The purpose of the work should emphasize the novelty of the research.

4. Results - very accurately described only climatic data. The results obtained for Harvested yield and quality characteristics are only presented in the table. For Protein, Ash, NDF, ADF, Crude Fiber only statistical relationships are given.

5 Discussion - The chapter is well described. However, it would be better if the reader could see the results.

6. Materials and Methods - please explain why such fertilization levels were given. Materials and Methods lacks described methods for determining yield quality characteristics measures.

7. Conclusion - the summary or conclusion should include the most important results obtained. In this work, the conclusion is very general.

8 References. In this work 52 items of literature were cited. Only 6 of them are works from the last 5 years. Please use more recent data.

Author Response

Title: "Could a legume-switchgrass sodseeding system increase forage productivity?"

Journal: Plants

Dear Editor,

We are thankful for giving us the opportunity to support our work. Following reviewers suggestion, we are sending the revised manuscript, based on our feeling that the work has been modified taking into account the suggestions made by the reviewers. In the revised manuscript, we have carried out important changes.

All the changes made in the revised manuscript are in colored yellow. For your decision and comparison with the first version, we present the changes in juxtaposition with reviewer’s suggestions in the next pages.

Sincerely yours,

Kyriakos Giannoulis

Checklist

  • The abstract of the work should include results and not just general statements.

Answer: Abstract has been corrected according to the comments.

  • The introduction needs to be revised, please provide relevant information to the topic of the work.

Answer: Introduction changed according to the comments.

  • The purpose of the work should emphasize the novelty of the research.

Answer: The novelty of the research is now clear.

  • Results - very accurately described only climatic data. The results obtained for Harvested yield and quality characteristics are only presented in the table. For Protein, Ash, NDF, ADF, Crude Fiber only statistical relationships are given.

Answer: Results changed according to the comments.

  • Discussion - The chapter is well described. However, it would be better if the reader could see the results.

Answer: Discussion changed according to the suggestions.

  • Materials and Methods - please explain why such fertilization levels were given. Materials and Methods lacks described methods for determining yield quality characteristics measures.

Answer: Materials and Methods have been checked and corrected according to the suggestions

  • Conclusion - the summary or conclusion should include the most important results obtained. In this work, the conclusion is very general.

Answer: Conclusion has been improved according to the comments

  • In this work 52 items of literature were cited. Only 6 of them are works from the last 5 years. Please use more recent data.

Answer: More recent data have been included

Round 2

Reviewer 1 Report

Dear colleagues,

I was glad to read all these very fruitful corrections for this interesting paper, however I still maintain the two points of my first review:

Point 3 although the photos 1 and 2 are welcome, we cannot see the plants at all as especially for photo 2 it is (only) general views, detailed photos are requested.

Point 4 As the mixture between results and discussion is still maintained and even increased with the new parts added (especially the very welcome paragraph “Finally, it would be interesting…”), in order to respect strictly each part of your text point 2 should be called “results and discussion” (notice also that in the present version there is “2.1”, but as there is no “2.2” your “.1” is not relevant!), then “point 3 discussion” should become point 2.2 and be called something like “general discussion”.

Author Response

Title: "Could a legume-switchgrass sodseeding system increase forage productivity?"

Journal: Plants

Following your suggestions, we are sending the revised manuscript, based on our feeling that the work has been modified taking into account the comments you made. All the changes made in the revised manuscript are colored yellow.

For your decision and comparison with the first version, we present the changes in juxtaposition in the next pages.

Sincerely yours,

Kyriakos Giannoulis

Checklist

  1. Point 3 although the photos 1 and 2 are welcome, we cannot see the plants at all as especially for photo 2 it is (only) general views, detailed photos are requested.

Answer: In photo 2 have been included photos where switchgrass, vetch and pea from the plots are clearly shown, according to your suggestions.   

  1. Point 4 As the mixture between results and discussion is still maintained and even increased with the new parts added (especially the very welcome paragraph “Finally, it would be interesting…”), in order to respect strictly each part of your text point 2 should be called “results and discussion” (notice also that in the present version there is “2.1”, but as there is no “2.2” your “.1” is not relevant!), then “point 3 discussion” should become point 2.2 and be called something like “general discussion”.

Answer: A typing mistake caused the numerical misunderstanding. Now the proper chapter numbers are included in the text according to your comments. Finally, “Discussion” is still the 3rd chapter as required by the template of the special issue. However, if you still consider that the numbering should be changed from 3 to 2.3 and the name from “Discussion” to “General Discussion”, will improve the article, then we could change it in the final draft.

Reviewer 2 Report

All my comments have been taken into account.

Author Response

Title: "Could a legume-switchgrass sodseeding system increase forage productivity?"

Journal: Plants

We are thankful for giving us the opportunity to support our work.

Sincerely yours,

Kyriakos Giannoulis

Checklist
All my comments have been taken into account
